# Adeno-Associated Virus Vectors in Retinal Gene Therapy: Challenges, Innovations, and Future Directions

**DOI:** 10.3390/biom15070940

**Published:** 2025-06-28

**Authors:** Jiayu Huang, Jiajun Li, Xiangzhong Xu, Keran Li

**Affiliations:** 1Department of Ophthalmology, The Affiliated Eye Hospital of Nanjing Medical University, 138 Hanzhong Road, Nanjing 210029, China; yellowhuangjiayu@gmail.com (J.H.); jiajunli0322@gmail.com (J.L.); 2The Fourth School of Clinical Medicine, Nanjing Medical University, Nanjing 210029, China

**Keywords:** adeno-associated virus (AAV), retinal gene therapy, capsid engineering, large-gene delivery, gene therapy-associated uveitis

## Abstract

Adeno-associated virus (AAV) vectors have emerged as the leading platform for retinal gene therapy due to their favorable safety profile, low immunogenicity, and ability to mediate long-term transgene expression within the immune-privileged ocular environment. By integrating diverse strategies such as gene augmentation and gene editing, AAV-based therapies have demonstrated considerable promise in treating both inherited and acquired retinal disorders. However, their clinical translation remains limited by several key challenges, including restricted packaging capacity, suboptimal transduction efficiency, the risk of gene therapy-associated uveitis, and broader societal concerns such as disease burden and ethical oversight. This review summarizes recent advances aimed at overcoming these barriers, with a particular focus on delivery route-specific disease applicability, multi-vector systems, and capsid engineering approaches to enhance payload capacity, targeting specificity, and biosafety. By synthesizing these developments, we propose a conceptual and technical framework for a more efficient, safer, and broadly applicable AAV platform to accelerate clinical adoption in retinal gene therapy.

## 1. Introduction

The blood–retinal barrier (BRB) confers a unique immune-privileged status upon the retina, creating a protective microenvironment. By limiting vector diffusion and systemic exposure, the BRB minimizes off-target effects and enhances the precision of retinal gene delivery. Coupled with advanced site-specific delivery techniques and real-time noninvasive imaging tools, the retina has emerged as an ideal target for gene therapy interventions [1].

Among available vectors, adeno-associated virus (AAV) has become the dominant platform for retinal gene delivery. Its non-pathogenic profile, low immunogenicity, and capacity for sustained transgene expression make AAV a highly suitable delivery vehicle. The structural plasticity of its capsid allows extensive optimization, and advances in vector engineering have markedly improved targeted transduction efficiency, further expanding its therapeutic potential [2].

Nonetheless, significant challenges remain across the continuum from basic research to clinical translation and global accessibility. Patients with inherited retinal disorders such as Usher syndrome and Stargardt disease often remain outside AAV’s current therapeutic window, underscoring the urgency of overcoming its limited cargo capacity [3,4]. The recent suspension of certain clinical trials has raised safety concerns related to dose-dependent inflammation and potential carcinogenicity, highlighting the critical need for robust safety assessment [5,6]. In addition, socioeconomic barriers, including patient financial burden and disparities in diagnostic and treatment infrastructure, further hinder the widespread implementation of AAV-based therapies.

This review provides a comprehensive analysis of the current landscape of AAV-mediated retinal gene therapy, including a comparative overview of delivery strategies. It emphasizes translational limitations while highlighting major technological advances and priority research directions. By integrating these perspectives, we aim to inform future investigations and accelerate the development of safe, effective, and accessible AAV-based treatments for retinal diseases.

## 2. AAV Vector Fundamentals: A Concise Toolkit for Retinal Therapy

### 2.1. AAV Structural Basis

AAV has become the first-line vector for retinal gene therapy due to its unique properties, with advantages and disadvantages compared to other viral vectors summarized in Figure 1.

Wild-type adeno-associated virus (wtAAV) is a replication-defective *dependoparvovirus* that requires co-infection with a helper virus for replication. In the absence of a helper, wtAAV can persist long-term in quiescent cells, enabling sustained transgene expression [7]. Although anti-AAV antibodies are prevalent in the population, no direct disease associations have been reported, supporting the clinical safety of AAV vectors [8].

wtAAV consists of a ~25 nm icosahedral capsid that encloses a ~4.7 kb single-stranded DNA (ssDNA) genome [9]. Its compact size facilitates retinal penetration but restricts the delivery of large or multi-gene constructs. The genome contains two essential genes: *rep* (replication) and *cap* (capsid). The *cap* gene encodes the three structural proteins VP1–3, whose sequence and conformational variations determine serotype tropism, forming the basis for capsid engineering [10]. In addition, the *cap* gene also encodes two nonstructural accessory proteins through alternative overlapping reading frames: assembly activating protein (AAP) and membrane-associated accessory protein (MAAP) [11]. AAP, derived from a +1 frame, is essential for capsid assembly by directing the nuclear localization and multimerization of VP proteins. MAAP, expressed from a +2 frame, contains a membrane-association domain and may facilitate AAV secretion via interaction with host trafficking pathways [12]. The inverted terminal repeats (ITRs) at both ends of the genome form hairpin structures that are essential for viral genome packaging and are also associated with vector immunogenicity [10]. To develop AAV as a gene therapy vector, recombinant AAV (rAAV) is constructed by replacing the native rep and cap genes with a therapeutic transgene cassette flanked by ITRs. In the absence of Rep expression, rAAV genomes predominantly persist as episomal forms within the host nucleus, minimizing integration-related risks. This design enhances safety, reduces immunogenicity, and maximizes available space for therapeutic payloads [13].

### 2.2. AAV Infection Process

Figure 2 illustrates the general infection pathway of common AAV serotypes [14,15]. After evading neutralizing antibodies, AAV initiates attachment via low-affinity interactions with cell surface glycans (e.g., heparan sulfate proteoglycans, HSPG), followed by high-affinity binding to serotype-specific receptors such as AAV receptor (AAVR) or, for certain serotypes like AAV5, PDGFR [16]. Internalization occurs primarily through clathrin-mediated endocytosis [17]. Notably, certain serotypes, such as AAV2, can also exploit the clathrin-independent carriers/GPI-enriched endocytic compartments (CLIC/GEEC) pathway, bypassing conventional trafficking and broadening cellular tropism to enhance transduction efficiency [18,19].

Following entry, some viral particles are degraded by the ubiquitin–proteasome system, while others escape into the cytoplasm, traffic along microtubules, and—under acidic conditions—undergo PLA2 domain exposure and activation [20]. This facilitates transport to the nucleus, where the viral genome is released through the nuclear pore complex into the host nucleus.

In this process, the efficiency of AAV-mediated transduction is influenced not only by the molecular interactions between the viral capsid and host cell surface receptors but also by the emerging role of the capsid in modulating the epigenetic landscape of the vector genome. Recent studies have demonstrated that specific AAV capsids can impact chromatin accessibility and histone modifications, thereby regulating transgene transcription. For instance, the AAV-LK03 capsid delivers genomes in murine cells lacking activating histone marks, leading to poor transcription. Inserting a single amino acid into the capsid restores these marks and significantly enhances expression [21]. Moreover, the VP1 N-terminal domain of the capsid has been shown to influence the chromatin state and transcriptional activity of the vector genome, underscoring the critical role of capsid structure in epigenetic regulation and transgene expression [22].

## 3. The Delivery Dilemma: Reconciling Safety, Efficacy, and Disease-Specific Applicability

Due to the unique anatomical and immunological features of the eye, the choice of delivery route for AAV-based gene therapy remains a critical determinant of both efficacy and safety. Among the various intraocular administration techniques, subretinal (SR) and intravitreal (IVT) injections are the most widely used in retinal clinical trials, yet they differ substantially in their technical demands, vector biodistribution, and translational applicability. This section provides a critical appraisal of SR and IVT delivery, highlighting their respective advantages, limitations, and disease-specific considerations, while also discussing emerging strategies that aim to overcome these challenges.

### 3.1. Subretinal Delivery: Precision at a Cost

SR delivery is widely regarded as the gold standard for targeting photoreceptors (PRs, including rods and cones) and retinal pigment epithelium (RPE) in inherited retinal diseases (IRDs). By delivering AAV vectors into the subretinal space between the neural retina and RPE, high-efficiency transduction can be achieved using relatively low vector doses. Several natural AAV serotypes, including AAV2/1, AAV2/2, AAV2/4, and AAV2/8, have shown strong tropism for PRs and RPE via this route [23].

One of the major advantages of SR delivery is its spatial precision and compartmentalization. The formation of a subretinal bleb creates a localized microenvironment that limits vector dissemination, reducing systemic exposure and mitigating immune activation [24]. This controlled environment also facilitates reliable dose–effect relationships, which are especially valuable in diseases with focal retinal degeneration.

However, this surgical route presents notable translational limitations. It requires pars plana vitrectomy, retinotomy, and intraoperative manipulation under a microscope—all of which carry risks of complications such as retinal detachment, macular hole formation, or iatrogenic photoreceptor damage. These risks are heightened in IRD patients with pre-existing retinal atrophy or thinning, particularly in the foveal region. Consequently, injections are often administered near the vascular arcades to avoid central macular trauma [25]. In addition, vector reflux from secondary bleb punctures may necessitate vitrectomy washout procedures, further extending surgical time [26,27].

Control of bleb height and tension is crucial [28]. Excessive detachment or pressure can cause transient ischemia, photoreceptor apoptosis, or chronic subretinal fluid accumulation. To mitigate this, pre-bleb formation using balanced salt solution (BSS) helps predict bleb spread and retain vector volume. Technological advances such as microscope-integrated OCT (MI-OCT), robotic-assisted microinjection systems (e.g., Preceyes BV), and non-vitrectomy subretinal delivery platforms (e.g., Orbit SDS™) have improved procedural accuracy and reduced variability among surgeons [29,30,31,32].

Even with surgical refinements, SR delivery’s limited area of transduction remains a concern, particularly for diseases requiring widespread retinal coverage [33]. Current strategies to overcome this limitation include AAV capsid optimization via directed evolution (screening high-efficiency variants in animal models) and rational design (modifying specific capsid residues based on structural insights) [34]. For example, AAV44.9, isolated from rhesus monkey kidney cell cultures, and its rationally designed variant AAV44.9 (E531D), have been shown to transduce over 98% of foveal cones and restore retinal function in mice, achieving distribution beyond conventional SR bleb boundaries [35]. Another engineered vector, mannose-conjugated rAAV2, retains native tropism while leveraging mannose-mediated uptake to significantly enhance transduction efficiency [36].

Despite these advances, SR delivery remains resource-intensive and is typically restricted to tertiary referral centers with vitreoretinal surgical capacity. These logistical barriers, coupled with surgical complexity, limit its scalability and routine clinical deployment, especially in low-resource settings or for chronic diseases requiring bilateral or repeat dosing.

### 3.2. Intravitreal Injection: Accessibility with Trade-Offs

IVT injection represents the most clinically accessible AAV delivery method. It is a quick, minimally invasive outpatient procedure that enables widespread vector distribution throughout the vitreous and inner retina [37]. Particularly for treating diseases affecting inner retinal layers or requiring pan-retinal expression, such as diabetic retinopathy (DR) or diabetic macular edema (DME), IVT provides practical advantages in terms of ease of repeat dosing, patient throughput, and cost-efficiency.

Preclinical studies have demonstrated that IVT delivery of AAV2 can transduce nearly all mouse retinal ganglion cells (RGCs), with AAV6 and AAV9 achieving partial Müller glial transduction [30]. However, efficient transduction of deeper retinal layers, especially PRs and RPE, remains a major hurdle. This is primarily due to the inner limiting membrane (ILM), a dense extracellular matrix that acts as a physical barrier to AAV diffusion. To overcome this, several engineered capsids have been developed. AAV2.7m8, created via peptide insertion at R587, enhances PR and RPE transduction in murine models and forms the basis of ADVM-022, which showed encouraging results in neovascular age-related macular degeneration (nAMD) trials without major safety concerns [38,39]. Nevertheless, the DME arm of the trial was terminated due to inflammation-induced vision loss, highlighting an ongoing challenge: the extent to which IVT-induced inflammation reflects capsid properties, delivery route, or disease-specific immune priming remains unresolved.

Next-generation engineered vectors, including AAV2.GL, AAV2.NN, and AAV2.100, aim to improve ILM penetration and cell-specific tropism. Notably, 4D Molecular Therapeutics’ AAV2.R100-based vectors have entered clinical trials. The 4D-125 preserved retinal architecture and improved photoreceptor function in X-linked retinitis pigmentosa (XLRP), while 4D-150 (co-expressing aflibercept and a VEGF-C-targeting RNAi) reduced disease burden by 83% in nAMD and decreased DME treatment frequency by 86%, alongside improved visual acuity and retinal thickness [40,41,42]. Some natural serotypes also show inherent ILM-penetrating ability. For instance, AAVrh10 (derived from rhesus macaques) can transduce outer photoreceptors via IVT, suggesting it naturally bypasses the ILM to reach deeper retinal layers [43].

Despite its translational appeal, IVT delivery is associated with a higher incidence of gene therapy-associated uveitis (GTAU). Meta-analyses report that 45% of IVT gene therapies are associated with GTAU, compared to 28% with SR delivery [44]. GTAU appears to be dose-dependent, with vector doses > 1 × 10^10^ vg/eye carrying increased risk. This has important implications for repeat dosing and re-administration, as neutralizing antibodies generated after the first dose may hinder subsequent treatments [45]. To address this, several immunomodulatory strategies have been developed. PEGylation of AAV capsids can shield antigenic sites without compromising transduction efficiency [46], while peptide insertion approaches (e.g., AAV2.MB453) disrupt toll-like receptor (TLR) signaling pathways [47]. Additionally, codon optimization and synthetic genome modifications can help evade TLR9-mediated innate immune responses [48,49].

Despite its limitations, IVT delivery remains the most scalable and patient-friendly approach. Its procedural simplicity and ability to support large-scale deployment make it an attractive platform for chronic diseases, particularly in aging populations or developing regions with limited access to retinal surgery.

### 3.3. Strategic Considerations: Disease Context Matters

A direct comparison between SR and IVT delivery underscores the importance of aligning the delivery strategy with the pathophysiology and anatomical target of the disease. For monogenic IRDs with confined retinal degeneration and well-defined gene targets, SR delivery remains the method of choice due to its precise targeting and high transduction efficiency. In contrast, for multifactorial diseases involving large retinal territories or requiring sustained anti-angiogenic expression, such as DR, IVT injection offers a more practical, scalable solution despite its immunological drawbacks.

Emerging methods such as suprachoroidal injection (SCI), which delivers vectors into the potential space between the sclera and choroid, offer a promising compromise. It bypasses the ILM and vitreous dilutional effects, enabling posterior segment targeting with reduced invasiveness [50]. Ongoing trials (e.g., ABBV-RGX-314 via SCI and SR) suggest comparable efficacy with improved tolerability [51,52]. Further head-to-head comparisons are needed to define the long-term therapeutic index and identify optimal use cases.

Ultimately, no single delivery route offers a universally optimal solution. Successful clinical translation of retinal gene therapies requires a context-specific and disease-informed strategy that takes into account anatomical targeting, vector immunogenicity, treatment durability, and the capacities of the healthcare system.

## 4. Inherited Retinal Diseases: AAV-Based Emerging Therapeutic Strategies

IRDs are a genetically and phenotypically heterogeneous group of retinal disorders characterized by progressive photoreceptor degeneration and functional impairment. This complexity limits the effectiveness of conventional treatments such as surgery and pharmacotherapy.

A major breakthrough came in 2017 with the approval of Luxturna, an AAV2-based therapy for biallelic RPE65-associated Leber congenital amaurosis (LCA), marking the first successful clinical translation of AAV in IRDs [53]. Since then, AAV has been widely applied in clinical trials for retinal gene therapy, many of which have shown encouraging preliminary results.

Depending on the nature of the mutation and its impact on retinal function, we systematically analyzed current IRD gene therapy clinical trials (Table 1) and classified existing strategies into three categories: gene augmentation to supplement functional genes, gene editing to correct mutant sequences, and RNA interference to regulate pathogenic gene expression (Figure 3). AAV-mediated RNA interference primarily uses shRNA to knock down target genes and will not be discussed in detail here [54].

These approaches aim to preserve or restore photoreceptors but are less effective in advanced disease with irreversible cell loss. In such cases, AAV-delivered optogenetics offers a complementary strategy by conferring light sensitivity to surviving inner retinal neurons, bypassing the need for functional photoreceptors.

### 4.1. AAV-Mediated Gene Augmentation Therapy

Gene augmentation therapy restores or compensates for lost or impaired gene function by introducing a functional gene into cells harboring pathogenic mutations, and is particularly suitable for autosomal recessive and X-linked disorders [55].

To date, most gene therapy candidates for IRDs have demonstrated favorable therapeutic outcomes in clinical studies [56]. Common endpoints in IRD trials include structural, functional, and patient-centered outcome measures. Unlike trials for acquired retinal diseases that often rely on best-corrected visual acuity (BCVA) or structural endpoints, IRD trials emphasize real-world functional improvements [57].

For instance, in conventional gene augmentation approaches, a systematic review of AAV-RPE65 gene therapy for LCA2 reported mild improvements in visual acuity, mobility, and full-field stimulus testing (FST) [58]. Similarly, a meta-analysis of retinitis pigmentosa (RP) showed enhanced light sensitivity and partial reversal of visual field loss [59].

Despite these advances, gene augmentation is limited to diseases with well-characterized mutations and genes small enough to fit within the ~4.7 kb packaging limit of AAV. Large genes such as ABCA4 (~6.8 kb), MYO7A (~7.5 kb), or USH2A (~15.7 kb) exceed this limit and require alternative delivery strategies.

### 4.2. AAV-Mediated Gene Editing

Gene editing enables the precise modification, deletion, or replacement of specific DNA sequences within target cells, directly correcting pathogenic mutations at their source. This approach has garnered significant attention in IRD gene therapy research and has demonstrated therapeutic potential that surpasses traditional gene augmentation strategies [60].

Among the available gene editing systems, CRISPR/Cas9 stands out for its high precision in recognizing and modifying specific genetic sequences. The system employs guide RNAs (gRNAs) to direct the Cas9 nuclease to the target site, where it induces double-strand breaks (DSBs). These breaks are then repaired by the cell via either homologous recombination (HDR) or non-homologous end joining (NHEJ), enabling the correction of disease-causing mutations [61].

HDR is particularly well suited for correcting monogenic mutations associated with IRDs and has been successfully applied to genes such as RPE65, RHO, and USH1C [62,63,64]. However, HDR is largely restricted to dividing cells, limiting its efficacy in terminally differentiated cells like photoreceptors and RPE cells. Furthermore, Cas9-induced DSBs pose a risk of insertion–deletion mutations (indels), which may compromise therapeutic outcomes. By contrast, NHEJ is more active in non-dividing cells and has shown promise for treating autosomal dominant RP. In a mouse model of the disease, NHEJ-based editing was shown to mitigate dominant-negative effects and partially restore photoreceptor function [65,66].

AAV-mediated gene editing is steadily advancing toward clinical translation [67]. The ongoing EDIT-101 trial utilizes AAV vectors to deliver SaCas9 and a gRNA targeting the CEP290 gene, aiming to excise a pathogenic intronic mutation and restore normal splicing in the treatment of LCA10 [68]. This represents the first clinical application of gene editing for a retinal disorder. Preliminary data suggest that CRISPR/Cas9-based editing is stable in vivo and exhibits a favorable safety profile.

Beyond conventional CRISPR/Cas9 editing, next-generation tools such as base editors (BEs) and prime editors (PEs) offer enhanced precision and reduced off-target effects [69,70]. Base editing employs a catalytically inactive Cas9 fused to cytosine or adenine deaminases, enabling single-nucleotide modifications without inducing DSBs. Prime editing combines a Cas9 nickase with a reverse transcriptase and uses a prime editing guide RNA (pegRNA) to introduce precise and minimally disruptive DNA edits [71,72,73].

Despite their immense potential, the clinical translation of these technologies is currently constrained by AAV delivery challenges [74]. The coding sequences of base and prime editors are too large to fit within a single AAV vector, limiting their feasibility. Overcoming these barriers will require ongoing innovation in vector engineering and delivery strategies to optimize gene editing platforms for retinal diseases [75].

### 4.3. AAV-Mediated Optogenetics

For patients with late-stage IRDs where photoreceptors are irreversibly lost, mutation-specific gene therapy is no longer feasible. Optogenetics offers a mutation-independent solution by introducing light-sensitive proteins into surviving inner retinal neurons (e.g., bipolar cells, RGCs), thereby bypassing the need for functional photoreceptors [76]. This approach has entered clinical evaluation. GS030, developed by GenSight Biologics, is based on an rAAV2.7m8 vector encoding the red-shifted channelrhodopsin ChrimsonR, fused to tdTomato for enhanced membrane targeting. Expression of ChrimsonR in central RGCs enables light-induced depolarization and action potential generation, effectively substituting for lost photoreceptor function. In an interim analysis involving nine RP patients, GS030 showed good safety and tolerability over 12 months, with no serious adverse events. Some participants improved from no light perception to being able to locate and count objects, demonstrating encouraging therapeutic potential [77,78]. However, nonselective activation of a broad RGC population may lead to ambiguous or distorted visual signals [79]. To address this, some studies have attempted to deliver distinct excitatory and inhibitory opsins to RGC somata and dendrites separately, aiming to improve signal fidelity [80].

In comparison, targeting bipolar cells activates the visual pathway earlier and is considered more physiologically relevant [81]. Preclinical studies in rd1 mice have shown that optogenetic stimulation of ON bipolar cells can partially restore both spatial and temporal visual responses. Clinically, MCO-010—a bipolar cell-targeted rAAV2-based therapy—has shown promising outcomes in the RESTORE trial: at week 76, 56% of participants maintained improved visual acuity without serious adverse events [82]. Nevertheless, due to the deep location of bipolar cells, achieving precise and cell-specific delivery remains challenging, requiring the development of more advanced delivery platforms.

For patients with advanced photoreceptor loss and undefined genetic mutations, optogenetics presents a promising therapeutic option. It lowers the reliance on genetic diagnostics, offering potential advantages for broader clinical applicability.

## 5. Large-Gene Delivery: Overcoming AAV Packaging Limits Through Advanced Vector Design

Many inherited retinal diseases are constrained by the limited AAV packaging capacity; several of the more common examples are summarized in Table 2.

Overcoming the inherent packaging limit of AAV vectors has become a central focus in expanding their therapeutic applicability. One of the most promising approaches involves multi-vector systems, in which oversized transgenes are divided into two or more fragments and packaged into separate AAV vectors for co-delivery. Within target cells, these fragments are reassembled through reconstruction mechanisms at the DNA, mRNA, or protein level, enabling full-length gene expression. Recent progress in dual and triple AAV strategies has extended the effective cargo capacity to approximately 9 kb and 14 kb, respectively, far exceeding the ~4.7 kb limitation of a single vector. This advancement enables the delivery of large genes, complex regulatory architectures, and genome editing platforms previously incompatible with AAV [83].

**Table 2 biomolecules-15-00940-t002:** Hereditary retinal diseases limited by large gene size for vector delivery.

Disease Name	Gene Name	cDNA Size (kb)	Affected Cell Types	Current Therapeutic Progress	Strategies to Overcome Size Limits
**Stargardt**	ABCA4	~6.8	PRs	**ABO-504**: Validated in murine/porcine models; preclinical phase [84] **VG-801**: High efficiency at transcript/protein levels [85] **OCU-410ST**: Phase I/II trial [86,87] **MCO-010**: Phase II trial [82,88] **SAR422459**: Phase I/II trial [89]	**1. Dual AAV systems** ABO-504: Intein-mediated dual-AAVVG-801: REVeRT technology **2. Non-gene replacement** OCU-410ST: AAV5-hRORA (encodes RORA) MCO-010: Optogenetic therapy **3. Alternative vectors** SAR422459: Lentiviral vector carrying ABCA4
**LCA10**	CEP290	~7.4	PRs (connecting cilium)	**EDIT-101:** Phase I/II clinical trial [68] **miniCEP290:** demonstrated efficacy in mouse models [90] **sepofarsen (QR-110):** Phase II/III clinical trial [91,92,93]	**1. In situ gene editing (CRISPR/Cas9)** EDIT-101: CRISPR-Cas9-based exon skipping/correction of the endogenous CEP290 gene in vivo **2. AON-mediated exon skipping** sepofarsen (QR-110): uses antisense oligonucleotides to restore correct CEP290 pre-mRNA splicing **3. Micro-gene construction** miniCEP290
**UsherI**	MYO7A	~6.7	PRs and inner ear hair cells	**AAVB-081:** Phase I/II clinical trial [94]	**1. Dual AAV vector constructs** AAVB-081: dual AAV8 delivering full-length MYO7A
	PCDH15	~5.7		**Dual AAV system**: demonstrated efficacy in mouse models [95] **miniPCDH15**: demonstrated efficacy in mouse models [96]	**1. Dual AAV system** DNA-level trans-splicing **2. Micro-gene construction** miniPCDH15
**Usher II**	USH2A	~15.7		**eSpCas9:** corrected USH2A mutations in iPSCs [97] **Ultevursen:** Phase II/III clinical trial [98,99] **pS/MAR-USH2A:** preliminary exploration in zebrafish models [100] **Minigene-4**: demonstrated efficacy in mouse models [101]	**1. Gene editing** eSpCas9: using CRISPR/eSpCas9-mediated genome editing to correct the two most common USH2A mutations in patient-derived iPSCs **2. AON-mediated exon skipping** Ultevursen: antisense oligonucleotide prevents aberrant USH2A pre-mRNA splicing **3. Use of non-AAV vectors** pS/MAR-USH2A: DNA plasmid

### 5.1. DNA Level

A key advantage of multi-vector delivery at the DNA level lies in its ability to reconstitute full-length genes within target cells through genomic recombination. This process relies primarily on three mechanisms: trans-splicing, homologous recombination, and hybrid approaches combining elements of both. Among these, the hybrid strategy has emerged as a preferred method, leveraging the advantages of both splicing and recombination to enhance reconstitution efficiency [102,103,104,105].

These strategies have not only demonstrated feasibility in experimental systems but have also achieved translational success in disease models. A representative example is the treatment of USH1B-associated retinitis pigmentosa caused by mutations in *MYO7A*. Researchers designed a dual AAV8 vector system carrying partial *MYO7A* sequences and incorporated both splicing and homologous arms to enable hybrid reconstruction. In both murine and non-human primate (NHP) models, this strategy achieved robust phenotypic rescue, validating the efficacy of high-efficiency multi-vector delivery [106]. Based on these findings, AAV-081, a dual AAV8 candidate, entered clinical trials. To date, no treatment-related adverse events or dose-limiting toxicities have been reported, and one participant experienced an improvement of more than one line in BCVA and over three lines in low-luminance visual acuity (LLVA), with positive trends observed in multiple functional endpoints [94]. These early outcomes provide compelling clinical support for the therapeutic potential of dual AAV platforms.

Despite these advances, DNA-level reconstitution raises several concerns [107]. First, variable protein expression may result from incomplete co-transduction of both vectors in the same cell. Second, truncated or misfolded proteins may be generated if recombination is incomplete, which can compromise efficacy and provoke immune responses.

Building upon the success of dual-vector approaches, triple AAV systems are being developed to further expand delivery capacity [108]. Though still in the early stages, these platforms have demonstrated the ability to reconstitute therapeutic genes exceeding 14 kb, offering a promising solution for large-gene IRDs with complex genetic architecture.

### 5.2. Protein Level

At the protein level, reconstruction depends on the use of split inteins—self-splicing peptide segments flanking the N- and C-termini of the therapeutic protein. Under physiological conditions, these inteins catalyze precise peptide ligation to restore the full-length protein in vivo. This strategy has been successfully applied to large genes such as *ABCA4* and *CEP290*, achieving expression levels in porcine retina models that exceed those of DNA-level dual AAV constructs and rival those of single-vector systems. These findings underscore the clinical potential of protein-level approaches in overcoming AAV’s packaging constraints [109,110].

Intein-mediated delivery is especially valuable for enabling AAV-compatible deployment of oversized gene-editing tools, including BEs and prime editors PEs. For example, cytosine or adenine base editors can be split into two parts and packaged separately. Upon co-delivery, intein-mediated reconstitution restores full enzymatic activity [111]. In LCA and RP models, AAV-intein-ABE systems have been shown to target RPE and photoreceptor cells, correct pathogenic mutations, and restore visual function, thereby accelerating the translational trajectory of base editing in ophthalmology [112,113].

To further enhance precision and versatility, PEs have also been adapted for AAV-intein delivery. Constructs incorporating Rma inteins (derived from *Rhodothermus marinus*) exhibit robust editing capabilities and broad applicability [114]. In RP models, AAV-delivered PEs successfully reversed photoreceptor loss and restored retinal function, providing strong preclinical support for PE application in IRDs [115].

Nonetheless, the AAV-intein platform presents some translational challenges. Residual intein peptides generated during protein splicing may trigger immune responses. To mitigate this, co-expression of Escherichia coli dihydrofolate reductase (ecDHFR) has been proposed to selectively degrade residual inteins and improve splicing fidelity [116]. Additionally, reconstitution efficiency depends heavily on intein type and cleavage site selection, both of which require extensive optimization. Continued efforts in intein engineering and site prediction will be critical for improving safety and efficacy in clinical settings [117].

### 5.3. mRNA Level

At the mRNA level, trans-splicing provides an alternative route for reconstituting large genes using dual AAVs. Unlike DNA- or protein-level methods, this approach operates post-transcriptionally, relying on the splicing of split mRNA fragments to generate functional full-length transcripts. A notable platform, REVeRT (RNA Editing via Trans-Splicing), has demonstrated this strategy’s potential. In a murine model of Stargardt disease, intravitreal delivery of REVeRT restored full-length *ABCA4* expression and partially rescued retinal structure and function [118].

Compared to DNA or protein-based reconstitution, mRNA trans-splicing is less constrained by specific splice sites and does not generate residual peptide byproducts, potentially reducing immunogenicity and offering greater design flexibility. Although still in early development, REVeRT and related systems significantly expand the AAV toolbox for large-gene delivery, offering a feasible and potentially safer alternative for treating IRDs involving oversized coding sequences.

### 5.4. Capacity Expansion Methods Other than Multiple AAV

In addition to multi-vector co-transduction, an increasingly utilized strategy involves engineering compact gene constructs that preserve essential functional domains. This includes truncated gene variants, miniaturized regulatory elements, and streamlined gene-editing tools.

For example, MiniCEP290 and Minigene-4, developed for LCA and USH2B, respectively, retain critical protein domains while remaining compatible with single-vector AAV delivery, avoiding the complexity of multi-vector systems [90,101].

Gene-editing tools such as SaCas9 (3.2 kb), a smaller ortholog of SpCas9, have been successfully packaged into a single AAV. In the EDIT-101 trial, SaCas9 and a CEP290-targeting gRNA were delivered via AAV5 to correct a deep intronic mutation in LCA10, demonstrating the feasibility of in vivo AAV-based gene editing [68,91].

While gene compaction strategies offer design flexibility, they may reduce expression levels or target specificity, requiring thorough validation. In contrast, multi-AAV systems remain the most robust and widely applicable approach for delivering large transgenes and are expected to play a central role in future gene therapies for oversized retinal disease targets.

## 6. Acquired Retinal Diseases: Novel Therapeutic Strategies for Multifactorial Pathogenesis

Acquired retinal diseases, such as DR and AMD, are major causes of vision loss worldwide. Unlike inherited retinal disorders caused by single-gene mutations, these conditions result from complex interactions between systemic and environmental factors, involving mechanisms such as inflammation, vascular dysfunction, and oxidative stress.

Current treatments, including intravitreal injections of anti-VEGF agents or corticosteroids, are limited by short duration and the need for frequent administration.

AAV-mediated gene therapy offers a promising alternative by enabling sustained intraocular expression of therapeutic proteins through a single injection. This is particularly advantageous for vascular-driven diseases like DR and AMD, potentially reducing treatment burden and improving long-term outcomes.

This section reviews recent clinical trials for acquired retinal diseases (Table 3) and summarizes the predominant therapeutic modalities of AAV-mediated gene therapy in this field (Figure 4).

### 6.1. Diabetic Retinopathy

DR is a prevalent microvascular complication of diabetes, classified into non-proliferative diabetic retinopathy (NPDR) and proliferative diabetic retinopathy (PDR). DME may occur at any stage of DR, manifesting as fluid accumulation and tissue swelling in the central foveal region [119].

Current AAV-mediated gene therapies for DR and DME primarily focus on sustained intraocular expression of anti-VEGF agents to suppress pathological neovascularization. Within clinical evaluation frameworks, improvement in the Diabetic Retinopathy Severity Scale (DRSS) remains the predominant primary endpoint [120]. However, recent trials increasingly incorporate composite or functional endpoints such as the “time to supplemental aflibercept injection,” which more directly capture the real-world utility of durable gene expression [121].

Among the most advanced clinical candidates, ABBV-RGX-314 utilizes an AAV8 vector to deliver a gene encoding an anti-VEGF monoclonal antibody fragment. In the Phase II ALTITUDE trial, 70.8% of patients receiving high-dose RGX-314 achieved a ≥1-step DRSS improvement, substantially outperforming the 25.0% response observed with conventional anti-VEGF injections. Moreover, RGX-314 reduced the risk of vision-threatening complications by 89% and demonstrated favorable tolerability [122]. Similarly, FT-003, a gene therapy product under clinical investigation in China and delivered via subretinal injection, has shown promising safety and therapeutic potential in preliminary studies [123].

Concurrently, preclinical investigations have explored alternative approaches to modulate the VEGF pathway through AAV-based gene delivery. Various strategies have been validated; for example, Flt23k and soluble Flt-1 (sFlt-1) neutralize VEGF by competitive binding or high-affinity antagonism, effectively suppressing abnormal vascularization [124,125]. These strategies underscore a conceptual shift from simple ligand blockade to pathway tuning, enabling tailored responses that may preserve physiological angiogenesis while inhibiting pathological neovascularization.

Emerging AAV-based interventions are increasingly designed to engage the broader pathogenic spectrum of DR, including inflammation, oxidative stress, and retinal neurodegeneration [126]. For instance, Müller glial cells, central to maintaining retinal homeostasis, have become a focal point for targeted interventions. AAV9 vectors driven by the GFAP promoter have enabled Müller cell-specific expression of the glucocorticoid receptor (GR), leading to restored visual function and neuroprotection in diabetic models [127]. Likewise, AAV-shH10-mediated restoration of Dp71 expression in Müller cells has been shown to reestablish BRB integrity, offering a potential therapeutic approach for DME [128].

These findings collectively underscore the robust delivery capacity and functional versatility of the AAV vector platform in addressing the multifactorial pathogenesis of DR. Furthermore, the integration of capsid engineering with cell-type-specific promoters provides an efficient and modular toolkit for dissecting the complex molecular mechanisms underlying retinal diseases [23]. Cell-type-specific promoters have now been established for nearly all major retinal cell types [129].

In sum, AAV-based gene therapy for DR is evolving beyond monolithic VEGF inhibition toward multi-pathway, cell-specific strategies that reflect the complex and progressive nature of the disease. This shift underscores a growing emphasis on patient-centered outcomes, not only anatomical or functional improvements, but also reduced treatment burden and long-term safety. Key to advancing this therapeutic paradigm is the integration of refined immunomodulation, precise capsid engineering, and promoter optimization to enhance delivery specificity and control gene expression dynamics. Equally important is the development of combinatorial vectors capable of targeting angiogenesis, neurodegeneration, and metabolic stress in parallel, mirroring the multifactorial pathology of DR. As these tools mature, AAV-based gene therapy holds the potential to shift diabetic retinopathy treatment from symptomatic management toward durable interventions that address underlying pathogenic mechanisms.

### 6.2. Age-Related Macular Degeneration

AMD is divided clinically into non-exudative (dry, dAMD) and exudative (neovascular, nAMD) forms. Although dAMD is more prevalent, nAMD causes the most severe vision loss due to choroidal neovascularization (CNV). Compared to DR or DME, nAMD responds better to anti-VEGF gene therapy, owing to localized CNV, clear RPE targets, and a more controlled inflammatory milieu.

ABBV-RGX-314 (AAV8-based) is under Phase I/II investigation for nAMD via SR and SCI delivery. In the SR cohort, 78% of patients required no supplemental anti-VEGF for nine months—equating to a 97% reduction in injection frequency—without corticosteroid prophylaxis. In a separate three-dose SCI study, RGX-314 was well tolerated, maintained best-corrected visual acuity, improved central retinal thickness at six months, and reduced annual injection burden by 80% [130,131].

Other AAV therapies for nAMD include Ixo-vec (formerly ADVM-022), an AAV2.7m8 vector encoding aflibercept. In the Phase I OPTIC trial, a single IVT injection reduced the annual anti-VEGF injection frequency in the fourth year by 86%, and intraocular aflibercept remained detectable for up to five years [132,133].

HG202—a CRISPR/Cas13 RNA-editing therapy delivered by a single AAV—marks the first clinical CRISPR RNA-editing approach for nAMD. In murine models, HG202 reduced CNV lesion area by 87%, outperforming both anti-VEGF antibodies and AAV-mediated anti-VEGF gene therapy. Editing VEGFA mRNA independently of receptor binding offers a novel mechanism for nAMD patients [134].

In contrast, dAMD features progressive PR and RPE loss, often resulting in geographic atrophy (GA), with no effective long-term treatments [135]. Complement inhibition is a major target in dAMD. GT005 (enhancing complement factor I) showed good safety but lacked efficacy [136]. The 4D-175 (AAV.R100-based) upregulates complement factor H, inhibiting complement activation and slowing degeneration, and has received IND clearance [137]. JNJ-1887 (AAV-CAG-sCD59) encodes soluble CD59 to block the membrane attack complex; Phase I trials in dAMD/nAMD reported good tolerability and no serious adverse events [138,139].

Beyond complement, non-angiogenic approaches are in development. OCU410 (AAV-mediated RORA expression) regulates lipid metabolism, reduces lipofuscin, and mitigates oxidative stress. Preclinical data support its safety and efficacy as a dAMD candidate [140].

From a translational standpoint, the therapeutic landscape of AMD offers several key insights for the development of retinal gene therapies. The relatively well-defined anatomical and immunological environment in AMD—particularly the localized pathology of nAMD and the confined target cell populations such as the RPE and choroidal endothelium—favors more predictable vector performance and greater therapeutic durability. In contrast, the diffuse microvascular and neuronal involvement in DR poses significant challenges for precise targeting and sustained efficacy. Furthermore, the modularity of AAV vectors, especially when coupled with emerging platforms like CRISPR/Cas13, enables a shift from traditional protein supplementation toward transcript-level modulation, allowing for highly tailored and potentially more durable interventions. Collectively, these features not only establish AMD as a leading candidate for gene therapy but also position it as an ideal model system for refining delivery routes, capsid engineering, and molecular strategies. Insights gained from ongoing AMD trials are therefore likely to inform future therapeutic approaches for more complex retinal diseases such as DR.

## 7. The Accessibility Frontier: From Bench to Bedside

While the scientific progress of AAV-based ocular gene therapy has been substantial, its successful clinical translation hinges not only on biological efficacy but also on broader factors such as cost, health system readiness, regulatory oversight, and ethical responsibility. Addressing these translational barriers is critical to ensuring equitable and sustainable access to retinal gene therapies on a global scale.

Despite growing public enthusiasm—surveys suggest over 90% of individuals express willingness to consider ocular gene therapy—only 45% demonstrate a clear understanding of its mechanisms and risks. Primary concerns include disease burden (38%), potential side effects (27%), and long-term safety [141]. These perceptions underscore the need for transparent, science-based public education campaigns and engagement strategies that can foster trust and informed decision-making.

Cost remains a central obstacle. The high price of Luxturna (~USD 850,000 for bilateral treatment) highlights the financial inaccessibility of gene therapies in current health systems. The economic burden is compounded by high vector manufacturing costs, stringent regulatory standards, and the relatively small patient populations targeted in ophthalmic applications. While some high-income countries may absorb these expenses through national insurance schemes or rare-disease funds, such therapies are often completely out of reach in low- and middle-income countries, exacerbating existing global health inequities [15].

Bridging these gaps will require scalable, cost-effective manufacturing solutions. Innovations in AI-guided vector optimization, high-throughput functional screening, and cell-free production systems are emerging as promising strategies to reduce production costs while maintaining consistency and safety standards [142,143]. Simultaneously, the broader clinical infrastructure must be strengthened. Gene diagnostics and advanced delivery techniques, such as subretinal injection, remain underutilized in primary care settings. Expanding access to genomic testing, establishing regional genetic screening centers, and training specialized personnel are essential to improve system readiness. Developing decentralized, multi-hub treatment centers may also provide the logistical support required for long-term implementation and follow-up.

Beyond technical and logistical considerations, ethical and regulatory dimensions are central to the responsible adoption of retinal gene therapy. For both AAV- and CRISPR-based interventions, robust ethical oversight is imperative. This includes systematic off-target effect monitoring, long-term safety surveillance, and oversight committees that involve not only clinicians and scientists but also patients and public health representatives. Furthermore, international harmonization of regulatory standards—particularly regarding genome-editing technologies—is essential to prevent unequal oversight and mitigate the risk of “ethics dumping” in jurisdictions with weaker regulatory frameworks.

In conclusion, the translation of retinal gene therapies from bench to bedside demands an integrated strategy—one that aligns scientific innovation with regulatory robustness, ethical accountability, and healthcare system adaptability. By proactively addressing these multidimensional challenges, gene therapy can evolve into a globally accessible and ethically grounded treatment paradigm.

## 8. Conclusions

As AAV-mediated retinal gene therapy advances beyond proof-of-concept and into broader clinical translation, the central question is no longer whether genes can be delivered, but how to architect therapeutic platforms that simultaneously achieve precision, persistence, and practicality. These three pillars form a dynamic and interdependent trilemma, in which optimizing one often comes at the expense of another. The future of the field depends not on maximizing a single attribute in isolation, but on achieving a context-specific, purpose-driven equilibrium among them.

Precision has evolved into a multi-dimensional construct encompassing molecular specificity, cell-type targeting, and anatomical delivery accuracy. Molecularly, the shift from gene augmentation to genome editing offers the potential for permanent correction of pathogenic variants. At the cellular level, the intersection of capsid engineering and cell-type-specific promoters has enabled increasingly selective transduction across distinct retinal neuronal populations. Anatomically, however, delivery remains a major constraint. SR injection enables high-precision delivery to photoreceptors and RPE, but at the cost of surgical complexity and limited accessibility. In contrast, IVT injection offers superior practicality but is hindered by diffusion barriers such as the ILM, compromising transduction depth and precision. Rather than selecting one approach over another, the path forward lies in the development of novel capsids, hybrid delivery routes (e.g., suprachoroidal or transscleral), and precision-guided surgical technologies that resolve, rather than merely shift, this trade-off.

Persistence, the defining promise of gene therapy, is increasingly challenged by immunogenicity and dose-related toxicity. The emergence of GTAU, the induction of neutralizing antibodies, and the loss of re-administration capacity all highlight the immunological fragility of current AAV platforms. Next-generation vectors must prioritize not only transduction efficiency but also immune evasion through capsid deimmunization, rational vector dose reduction, and prophylactic immune modulation strategies. These approaches are not peripheral enhancements; they are foundational to achieving durable therapeutic outcomes, especially in chronic or progressive retinal diseases.

Practicality—often underappreciated in early-phase innovation—is emerging as a principal constraint on widespread clinical impact. Biologically optimal vectors are of limited value if they cannot be produced at scale, distributed efficiently, or administered within real-world healthcare systems. As the focus shifts from rare IRDs to prevalent diseases such as DR and AMD, practicality becomes a central axis of therapeutic design. The high costs of manufacturing, delivery infrastructure, and specialized surgical requirements risk exacerbating disparities in access. Meeting this challenge will require not only continued advances in vector design but also streamlined manufacturing, scalable delivery methods, and supportive regulatory and reimbursement frameworks.

In synthesis, the future of AAV-mediated retinal gene therapy lies in navigating—not eliminating—the trilemma between precision, persistence, and practicality. These forces exist in dynamic tension: maximizing anatomical precision via SR delivery may reduce scalability; increasing vector dose to enhance persistence may provoke immune rejection; prioritizing minimally invasive routes may limit cellular targeting. The field is thus transitioning from a vector-centric to a disease-contextualized therapeutic model. For monogenic IRDs in young patients, maximizing precision and durability may be appropriate; for widespread, chronic conditions, practicality and immunological tolerance will be paramount.

Ultimately, success will be defined not by any single universal solution but by the development of a modular, programmable AAV platform that enables rational tailoring of vector properties, delivery strategies, and immunological safeguards to each disease context. Such a toolkit will mark the true maturation of retinal gene therapy: not merely as a technological innovation, but as a sustainable, scalable, and globally accessible therapeutic modality.

## Figures and Tables

**Figure 1 biomolecules-15-00940-f001:**
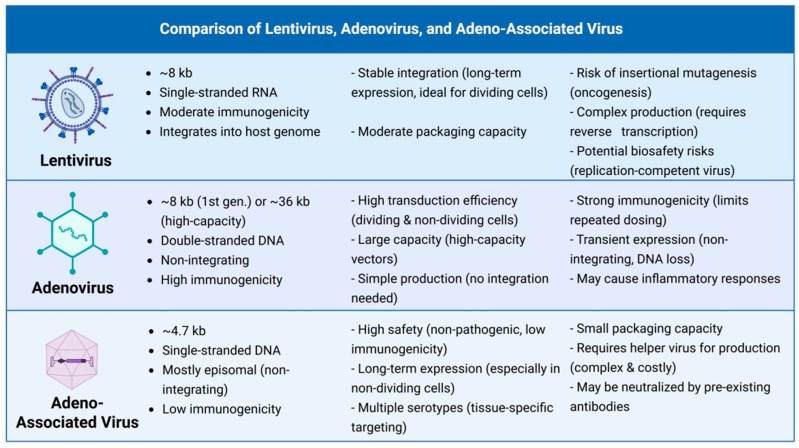
Comparison of Lentivirus, Adenovirus, and Adeno-Associated Virus. Created with BioRender (http://app.biorender.com), accessed on 22 April 2025.

**Figure 2 biomolecules-15-00940-f002:**
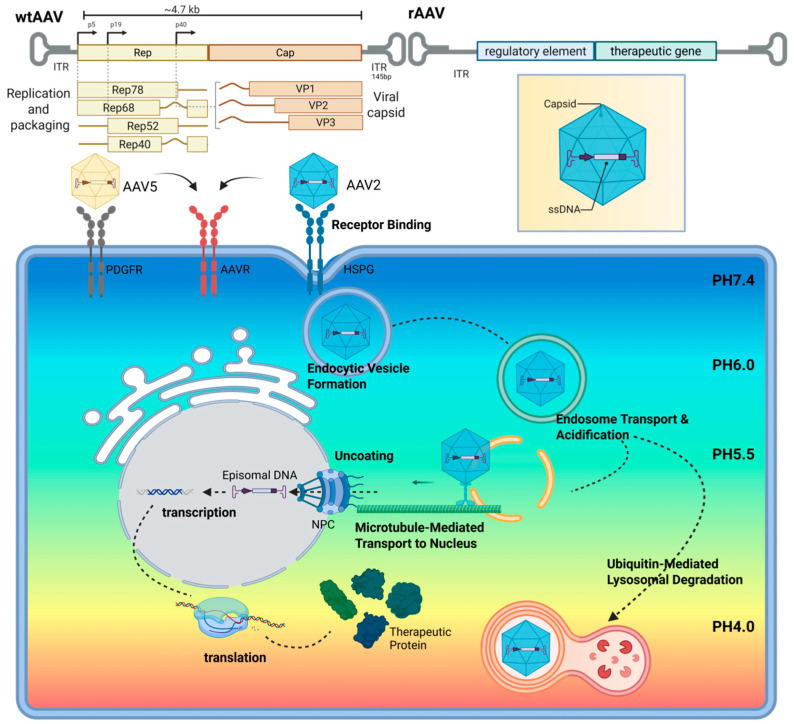
Schematic of the general process of AAV infection in target cells. Created with BioRender (http://app.biorender.com), accessed on 22 April 2025.

**Figure 3 biomolecules-15-00940-f003:**
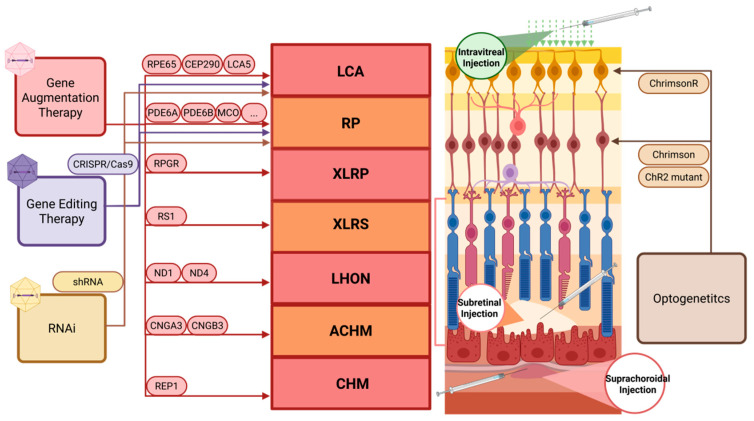
Application of AAV-mediated gene therapy in hereditary retinal diseases. Created with BioRender (http://app.biorender.com), accessed on 22 April 2025. Color-coded arrows indicate: Red arrows: Gene augmentation therapy; Purple arrows: Gene editing therapy; Yellow arrows: RNA interference (RNAi) therapy; Brown arrows: Optogenetic therapy.

**Figure 4 biomolecules-15-00940-f004:**
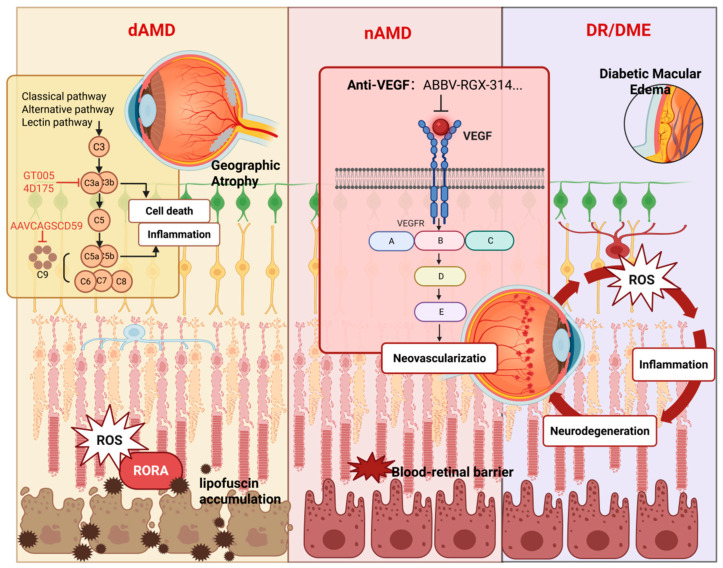
Application of AAV-mediated gene therapy in acquired retinal diseases. Created with BioRender (http://app.biorender.com), accessed on 22 April 2025.

**Table 1 biomolecules-15-00940-t001:** Clinical trials based on AAV vectors for hereditary retinal diseases.

Disease	Major Affected Cell Types	Therapeutic Target Cell Types	Target Genes	Sponsor	Phase	Interventions and Vector	Delivery	NCT number
**RPE65 Mutation-associated Retinal Dystrophy**	RPEs; secondary PRs	RPEs	RPE65	Novartis Pharmaceuticals	III	Voretigene neparvovec-rzyl (AAV2)	SR	NCT04516369
				Innostellar Biotherapeutics Co.	I/II	LX101 (AAV2)	SR	NCT06196827
								NCT06212297
				Frontera Therapeutics	I/II	FT-001 (AAV2)	SR	NCT05858983
**LCA**	PRs; RPEs		RPE65	Spark Therapeutics	I	Voretigene neparvovec-rzyl (rAAV2)	SR	NCT00516477
								NCT01208389
					III			NCT01208389
				Applied Genetic Technologies Corp	I/II	rAAV2-CB-hRPE65	SR	NCT00749957
				MeiraGTx UK II Ltd.	I/II	MGT003 (rAAV2)	SR	NCT02781480
				Hadassah Medical Organization	I	rAAV2-hRPE65	SR	NCT00821340
				Nantes University Hospital	I/II	rAAV2/4.hRPE65	SR	NCT01496040
				University of Pennsylvania	I	rAAV2-CBSB-hRPE65	SR	NCT00481546
				HuidaGene Therapeutics Co.	I/II	HG004 (rAAV9)	SR	NCT05906953
				University College, London	I/II	tgAAG76 (rAAV2/2)	SR	NCT00643747
		PRs	CEP290	Ocugen	I/II	OCU400 (rAAV5)	SR	NCT05203939
			CEP290	Editas Medicine	I/II	EDIT-101 (rAAV5)	SR	NCT03872479
			GUCY2D	Atsena Therapeutics	I/II	ATSN-101 (rAAV5)	SR	NCT03920007
**RP**	Rods, secondary Cones and RPEs	PRs	PDE6A	STZ eyetrial	I/II	Raav8.hPDE6A	SR	NCT04611503
			PDE6B	eyeDNA Therapeutics	I/II	rAAV2/5-hPDE6B	SR	NCT03328130
		Cones	RHO, PDE6A or PDE6B	SparingVision	I/II	SPVN06 (rAAV)	SR	NCT05748873
		RPEs	RLBP1	Novartis Pharmaceuticals	I/II	CPK850 (rAAV8)	SR	NCT03374657
		Bipolar cells	MCO	Nanoscope Therapeutics Inc.	I/II	vMCO-I (rAAV2)	IVT	NCT04919473
					II	MCO-010 (rAAV2)	IVT	NCT04945772
		RGCs	ChrimsonR	GenSight Biologics	I/II	GS030 (rAAV2.7m8)	IVT	NCT03326336
**XLRP**	Rods, secondary Cones and RPEs	PRs	RPGR	Beacon Therapeutics	I/II	AGTC-501 (rAAV2tYF)	SR	NCT03316560
					II			NCT06333249
					II			NCT06275620
					II/III			NCT04850118
				Janssen Pharmaceutical K.K.	II	rAAV2/5-hRKp.RPGR	SR	NCT06646289
					III			NCT04671433
								NCT04794101
								NCT05926583
				Biogen (NightstaRx Ltd., a Biogen Company)	I/II/III	BIIB112 (rAAV8-RPGR)	SR	NCT03116113
					III	BIIB111 (rAAV2-REP1); BIIB112 (rAAV8-RPGR)	SR	NCT03584165
				MeiraGTx UK II Ltd.	I/II	rAAV2/5-RPGR	SR	NCT03252847
				Frontera Therapeutics	I/II	FT-002 (rAAV)	SR	NCT06492850
				4D Molecular Therapeutics	I/II	4D-125 (rAAV.R100)	IVT	NCT04517149
**XLRS**	PRs; Bipolar cells	PRs; Bipolar cells	RS1	Atsena Therapeutics Inc.	I/II	ATSN-201 (AAV.SPR-hGRK1-hRS1syn)	SR	NCT05878860
				Applied Genetic Technologies Corp	I/II	rAAV2tYF-CB-hRS1	IVT	NCT02416622
				VegaVect	I/II	rAAV8-scRS/IRBPhRS	IVT	NCT02317887
**LHON**	RGCs	RGCs	ND1	Neurophth Therapeutics Inc.	I/II	NFS-02 (rAAV2-ND1)	IVT	NCT05820152
			ND4	Neurophth Therapeutics Inc.	I/II	NR082 (rAAV-ND4)	IVT	NCT05293626
				Wuhan Neurophth Biotechnology Limited Company	II/III			NCT04912843
				GenSight Biologics	I/II	GS010 (rAAV2/2-ND4)	IVT	NCT02064569
					III			NCT03406104
								NCT03293524
								NCT02652780
								NCT02652767
				Huazhong University of Science and Technology	II/III	rAAV2-ND4	IVT	NCT03153293
				Byron Lam	I	scAAV2-P1ND4v2	IVT	NCT02161380
**ACHM**	Cones	Cones	CNGA3	STZ eyetrial	I/II	rAAV.hCNGA3	SR	NCT02610582
				MeiraGTx UK II Ltd.	I/II	rAAV2/8-hG1.7p.coCNGA3	SR	NCT03758404
			CNGB3		I/II	rAAV2/8-hG1.7p.coCNGB3	SR	NCT03001310
			CNGA3 and CNGB3		I/II	rAAV2/8-hCARp.hCNGB3 and rAAV2/8-hG1.7p.coCNGA3	SR	NCT03278873
			CNGB3	Applied Genetic Technologies Corp	I/II	rAAV2tYF-PR1.7-hCNGB3	SR	NCT02599922
**CHM**	RPEs; secondary PRs	RPEs	REP1	Biogen	II	BIIB111 (rAAV2-REP1)	SR	NCT03507686
					III	BIIB111 (rAAV2-REP1)	SR	NCT03496012
				Biogen (NightstaRx Ltd., a Biogen Company)	III	BIIB111 (rAAV2-REP1); BIIB112 (AAV8-RPGR)	SR	NCT03584165
				University of Oxford	I/II	rAAV2.REP1	SR	NCT01461213
					II	rAAV2.REP1	SR	NCT02407678
				Byron Lam	II	rAAV2-REP1	SR	NCT02553135
				University of Alberta	I/II	rAAV2.REP1	SR	NCT02077361
				STZ eyetrial	II	rAAV2.REP1	SR	NCT02671539
				Spark Therapeutics	I/II	rAAV2-hCHM	SR	NCT02341807
				4D Molecular Therapeutics	I	4D-110 (rAAV.R100)	IVT	NCT04483440

**Table 3 biomolecules-15-00940-t003:** Clinical trials based on AAV vectors for acquired retinal diseases.

Mechanism of Action	Interventions and Vector	Sponsor	Disease	Phase	Delivery	NCT number
**Expressing aflibercept**	ADVM-022 (AAV.7m8)	Adverum Biotechnologies	DME	II	IVT	NCT05607810
			DME	II	IVT	NCT04418427
			nAMD	I	IVT	NCT03748784
			nAMD	II	IVT	NCT05536973
			nAMD	II	IVT	NCT04645212
**Expressing both aflibercept and a VEGF-C inhibitory RNAi**	4D-150(AAV.R100)	4D Molecular Therapeutics	DME	II	IVT	NCT05930561
			nAMD	I/II	IVT	NCT05197270
**Encoding a ranibizumab-like anti-VEGF monoclonal antibody fragment**	ABBV-RGX-314 (AAV8)	REGENXBIO Inc.	nAMD	I/II	SR	NCT03066258
		AbbVie	DME	II	SCS	NCT04567550
			nAMD	II	SR	NCT04832724
			nAMD	II	SR	NCT03999801
			nAMD	II/III	SR	NCT04704921
			nAMD	III	SR	NCT05407636
			nAMD	II	SCS	NCT04514653
**Encoding a confidential anti-VEGF protein**	FT-003	Frontera Therapeutics	DME	I	IVT	NCT05916391
			DME	I/II	IVT	NCT06492876
			nAMD	I	IVT	NCT05611424
			nAMD	I/II	IVT	NCT06492863
**Encoding a confidential anti-VEGF protein**	SKG0106	Skyline Therapeutics (US) Inc.	nAMD	I/II	IVT	NCT05986864
		Wang Min	DME	I	IVT	NCT06237777
		Youxin Chen	nAMD	I	IVT	NCT06213038
**Encoding a confidential anti-VEGF protein**	KH658	Chengdu Origen Biotechnology Co.	nAMD	I	SCS	NCT05657301
**Expressing human VEGF receptor fusion protein with binding affinity to VEGF-A, VEGF-B, and PlGF**	KH631		nAMD	I/II	SR	NCT05672121
**Encoding a confidential anti-VEGF protein**	LX102	Innostellar Biotherapeutics Co.	nAMD	I	SR	NCT06198413
			nAMD	II	SR	NCT06196840
**Encoding a confidential anti-VEGF protein**	RRG001	Shanghai Refreshgene Technology Co.	nAMD	I/II	SR	NCT06141460
**Encoding a soluble VEGF receptor**	rAAV.sFlt-1	Lions Eye Institute, Perth, Western Australia	nAMD	I/II	SR	NCT01494805
	AAV2-sFLT01	Genzyme, a Sanofi Company	nAMD	I	IVT	NCT01024998
**Encoding an angiopoietin domain and VEGF receptor (ABD-VEGFR) fusion protein**	EXG102-031	Exegenesis Bio	nAMD	I	SR	NCT05903794
**Targeted degradation of retinal VEGF-A mRNA via high-fidelity CRISPR/Cas13Y (hfCas13Y) system.**	HG202	HuidaGene	nAMD	I	SR	NCT06623279
**Complement pathway modulation: encoding a soluble form of CD59 (sCD59)**	JNJ-1887 (AAV2)	Janssen Research and Development, LLC	nAMD	I	IVT	NCT03585556
			dAMD	II	IVT	NCT05811351
				I	IVT	NCT03144999
				II	IVT	NCT06635148
				II	IVT	NCT04358471
**Complement pathway modulation: encoding encoding human CFI**	GT005 (AAV2)	Gyroscope Therapeutics Limited	dAMD	II	SR	NCT05481827
**Anti-inflammatory: encoding RORα**	OCU410 (AAV5)	Ocugen	dAMD	I/II	SR	NCT06018558

## Data Availability

Not applicable.

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
