# Peer review of "Adeno-Associated Virus Vectors in Retinal Gene Therapy: Challenges, Innovations, and Future Directions"

_biomolecules, 2025, doi:10.3390/biom15070940_

Round 1

Reviewer 1 Report

Comments and Suggestions for Authors

Huang and colleagues present a comprehensive review on AAV vector mediated gene therapy of inherited and acquired retinal diseases. They discuss current clinical trials as well as strategies to overcome current limitations, including limited transgene capacity, delivery precision, and immunogeneicity, in great detail. The manuscript is well written and easy to follow, although it would profit from reducing the length. The illustrations are informative.

Minor comments:

  1. Line 84: replace dependovirus by dependoparvovirus
  2. Line 102-105: mention AAP and MAAP for completedness
  3. Lines 121-125: mention CLIC/GEEC pathway and include reference: Nonnenmacher and Weber, 2011 (DOI: 10.1016/j.chom.2011.10.0149)
  4. The length should be somewhat reduced.

Author Response

Dear Reviewer,

Thank you very much for your positive and detailed review of our manuscript titled "Adeno Associated Virus Vectors in Retinal Gene Therapy: Challenges, Innovations, and Future Directions." We sincerely appreciate your recognition of the clarity, structure, and thoroughness of our review, as well as your kind comments on the quality of the illustrations. We also value your constructive suggestion regarding the manuscript length. In response, we have carefully revised the text to enhance conciseness while preserving the depth and clarity of the content. Your insightful feedback has been highly motivating and instrumental in improving the overall quality of our work. Below are our point-by-point responses to the reviewers' comments.

Comment No.1:

Line 84: replace dependovirus by dependoparvovirus

Response:

Thank you for your valuable comments. We have carefully reviewed, verified, and revised the relevant sections accordingly. To ensure clarity regarding the changes made, all modifications have been clearly highlighted in the revised manuscript. This revision can be found in lines 85-86 of the revised manuscript.

Comment No.2:

Line 102-105: mention AAP and MAAP for completeness

Response:

Thank you for your valuable comments. We have conducted an in-depth review of AAP and MAAP. The inclusion of this information has enhanced the section on AAV structural biology and provides a foundational understanding for capsid engineering and vector production. The added content has been clearly highlighted in the revised manuscript. This revision can be found in lines 95-101 of the revised manuscript.

Comment No.3:

Lines 121-125: mention CLIC/GEEC pathway and include reference: Nonnenmacher and Weber, 2011 (DOI: 10.1016/j.chom.2011.10.0149)

Response:

Thank you for your valuable comments. The reference you recommended has enriched our discussion of the endocytic process of AAV and provided additional insight into its infection pathway. We have incorporated a description of how AAV2 enters target cells via the CLIC/GEEC pathway in the relevant section. This supplementary content has been clearly highlighted in the revised manuscript. This revision can be found in lines 114-118 of the revised manuscript.

Comment No.4:

The length should be somewhat reduced.

Response:

Thank you for your valuable comments. We have restructured the overall framework of the review, shortened its length, and placed greater emphasis on the comparative analysis of different AAV delivery strategies for retinal diseases, as well as the key obstacles encountered during clinical translation. By focusing on practical challenges in translational application and incorporating a discussion of relevant societal factors, we aimed to make the review more aligned with the needs and perspectives of clinical practitioners.

We sincerely thank you for your careful and constructive review of our manuscript. Your insightful comments and suggestions have greatly contributed to improving the clarity, depth, and overall quality of our work. We have addressed each point thoroughly and believe the revisions have significantly strengthened the manuscript. We greatly appreciate your time and expertise, and look forward to any further feedback you may have.

Reviewer 2 Report

Comments and Suggestions for Authors

Huang et al presented a comprehensive review on AAV-based retinal gene therapy. The content is informative and well organized, with illustrations demonstrating the key concepts and tables summarizing the current landscape in the field. This reviewer has no major concerns. A few suggestions are as follows:

  1. Lines 130-137: In addition to capsid-receptor interactions, an emerging aspect of AAV-mediated transgene expression is about the interaction between the capsid and transgene transcription, including some capsids impacting the epigenetics of the vector genome. These literature should be discussed, e.g., PMID 37117181, 38431840.
  2. Table 1: It will be informative to also indicate the major cell type(s) associated with each disease, e.g., rod, cone, RPE, etc. If the gene therapy is designed to target a different cell type, it should also be listed. These information will provide guidance to develop gene therapies targeting other ocular diseases afflicting these cell types.
  3. Section 4.1: It will be helpful to generate a table summarizing the diseases where large genes exceeding AAV cargo limit need to be delivered. It may include disease names, gene names, gene (cDNA) sizes, cell types impacted, any current gene therapy development and strategies to overcome the size limit. It will help readers to quickly grasp the landscape around this topic.
  4. Lines 675-680: The description on AAV44.9 is not accurate. Please refer to PMID 32304666 and better describe the origine of this AAV capsid.

Author Response

Dear Reviewer,

Thank you very much for your positive and encouraging comments on our manuscript titled "Adeno-Associated Virus Vectors in Retinal Gene Therapy: Challenges, Innovations, and Future Directions." We are grateful for your recognition of the manuscript's informativeness, organization, and the clarity provided by the illustrations and tables. We appreciate your constructive suggestions and will carefully consider them to further improve the quality of our work.

Comment No.1:

Lines 130-137: In addition to capsid-receptor interactions, an emerging aspect of AAV-mediated transgene expression is about the interaction between the capsid and transgene transcription, including some capsids impacting the epigenetics of the vector genome. These literature should be discussed, e.g., PMID 37117181, 38431840.

Response:

Thank you for your valuable comments. The reference you recommended provides a thorough analysis of the interplay between the AAV capsid and transgene expression, offering strong theoretical support for our discussion on capsid optimization by elucidating its interactions with both target cell receptors and transgene transcription processes. Accordingly, we have revised and expanded the relevant sections, with all changes clearly indicated in the updated manuscript. This revision can be found in lines 126-136 of the revised manuscript.

Comment No.2:

Table 1: It will be informative to also indicate the major cell type(s) associated with each disease, e.g., rod, cone, RPE, etc. If the gene therapy is designed to target a different cell type, it should also be listed. These information will provide guidance to develop gene therapies targeting other ocular diseases afflicting these cell types.

Response:

Thank you for your valuable comments. In response, we have added two new columns to Table 1 to indicate both the major retinal cell types affected by each disease and the intended target cells of the corresponding gene therapy. We believe these additions improve the clarity and translational relevance of the table, offering a valuable reference for the development of gene therapies targeting similar cell types or disease mechanisms. This revision can be found in lines 276-277 of the revised manuscript.

Comment No.3:

Section 4.1: It will be helpful to generate a table summarizing the diseases where large genes exceeding AAV cargo limit need to be delivered. It may include disease names, gene names, gene (cDNA) sizes, cell types impacted, any current gene therapy development and strategies to overcome the size limit. It will help readers to quickly grasp the landscape around this topic.

Response:

Thank you for your valuable comments. We have reviewed the target cell types commonly addressed in gene therapies for inherited retinal diseases in clinical settings. These target cells are primarily selected based on the specific cell types affected by each disease. In response to your suggestion, we have added a new table (Table 2), with the aim of providing a useful reference for the development of gene therapies targeting similar cell types or disease categories. This revision can be found in lines 375-376 of the revised manuscript.

Comment No.4:

Lines 675-680: The description on AAV44.9 is not accurate. Please refer to PMID 32304666 and better describe the origine of this AAV capsid.

Response:

Thank you for your valuable comments. We have carefully reviewed the reference you recommended and examined additional literature related to AAV44.9. Based on this, we have corrected the description regarding the origin of AAV44.9. The revised content has been clearly indicated in the submitted manuscript. This revision can be found in lines 178-182 of the revised manuscript.

We sincerely appreciate the time and effort you have devoted to reviewing our manuscript and providing such insightful and constructive feedback. Your comments have been invaluable in enhancing the clarity, rigor, and overall quality of our work. We believe that the revisions we have made in response to your suggestions have significantly strengthened the manuscript. We look forward to any further guidance you may have and thank you once again for your thoughtful evaluation.

Reviewer 3 Report

Comments and Suggestions for Authors

Dear Authors,

It is a pleasure to review your manuscript “"Adeno-Associated Virus Vectors in Retinal Gene Therapy: Challenges, Innovations, and Future Directions". The review provides a comprehensive and well-structured overview of adeno-associated virus (AAV)-based retinal gene therapy, effectively synthesizing the current state of the field. The authors adeptly cover the physiological basis of AAV, its applications in inherited and acquired retinal diseases, and key challenges such as limited packaging capacity, delivery barriers, and immunogenicity.

The inclusion of detailed tables summarizing clinical trials (Tables 1 and 2) and illustrative figures enhances the manuscript’s utility as a valuable reference for researchers and clinicians. The discussion of recent innovations, such as exosome-associated AAVs and capsid engineering, reflects engagement with cutting-edge developments, and the clear organization facilitates accessibility for a broad audience.

However, the review’s novelty could be strengthened to distinguish it from existing literature, such as:

  • Ail, D., Malki, H., Zin, E. A., & Dalkara, D. (2023). Adeno-Associated Virus (AAV) - Based Gene Therapies for Retinal Diseases: Where are We?.The application of clinical genetics16, 111–130. https://doi.org/10.2147/TACG.S383453
  • Mendell, J. R., Al-Zaidy, S. A., Rodino-Klapac, L. R., Goodspeed, K., Gray, S. J., Kay, C. N., Boye, S. L., Boye, S. E., George, L. A., Salabarria, S., Corti, M., Byrne, B. J., & Tremblay, J. P. (2021). Current Clinical Applications of In Vivo Gene Therapy with AAVs.Molecular therapy : the journal of the American Society of Gene Therapy29(2), 464–488. https://doi.org/10.1016/j.ymthe.2020.12.007
  • Bulcha, J.T., Wang, Y., Ma, H.et al. Viral vector platforms within the gene therapy landscape. Sig Transduct Target Ther 6, 53 (2021). https://doi.org/10.1038/s41392-021-00487-6

which similar themes, including AAV vector design, delivery methods, and clinical applications. Although the current manuscript presents this information in an accessible and well-structured manner, it remains largely descriptive and does not offer a sufficiently distinctive analytical perspective.

To enhance the manuscript’s originality and impact, the authors could consider incorporating a more critical comparison of delivery strategies, such as subretinal versus intravitreal injection, evaluating their respective translational hurdles and applicability to different disease contexts.

Additionally, expanding the discussion to include patient-centric outcomes could broaden the manuscript’s relevance, particularly for clinicians, healthcare providers, and policy stakeholders.

The inclusion of commentary on ethical and regulatory aspects of retinal gene therapy, or disparities in access to clinical trials globally, would further enrich the manuscript.

Author Response

Dear Reviewer,

Thank you very much for your positive evaluation and detailed review of our manuscript titled "Adeno-Associated Virus Vectors in Retinal Gene Therapy: Challenges, Innovations, and Future Directions." We sincerely appreciate your recognition of the manuscript's structure, clarity, and its value as a reference for both researchers and clinicians. We are also grateful for your constructive suggestions, which have been instrumental in enhancing the originality, analytical depth, and translational relevance of our review.

We have carefully considered and addressed each of your comments, as detailed below:

Comment No.1:

Although the current manuscript presents this information in an accessible and well-structured manner, it remains largely descriptive and does not offer a sufficiently distinctive analytical perspective.

Response:

Thank you for your valuable comments. We have revised the relevant sections and adjusted the focus of the manuscript to strengthen the analytical perspective. In the updated overview of current clinical trials, we emphasize the conceptual shift in DR treatment strategies from single-pathway VEGF inhibition to multi-target, cell-specific approaches, as well as the anatomical and immunological advantages of AMD as a model for developing gene therapies for complex retinal diseases. We also discuss key translational priorities for AAV-based therapies, including immunomodulation, capsid/promoter engineering, and combinatorial gene delivery. Furthermore, we highlight the importance of patient-centered clinical outcomes beyond anatomical improvements to enhance the manuscript's clinical relevance.

In addition, we have substantially revised the "Conclusions" section to enhance its analytical depth and originality. Specifically, we introduce the conceptual framework of a "trilemma" among precision, persistence, and practicality as a central issue in the future development of AAV-mediated retinal gene therapy. Building on the existing content, this section offers independent reflections on how to balance these competing priorities across different disease contexts. We believe this revision significantly strengthens the manuscript's depth, originality, and translational relevance.

This revision can be found in lines 528–539, 573586, and 629681 of the revised manuscript.

Comment No.2:

To enhance the manuscript's originality and impact, the authors could consider incorporating a more critical comparison of delivery strategies, such as subretinal versus intravitreal injection, evaluating their respective translational hurdles and applicability to different disease contexts.

Response:

Thank you for your valuable comments. We fully agree with your recommendation to strengthen the comparative discussion of delivery methods. In the revised manuscript, we have expanded the relevant section to systematically compare subretinal and intravitreal injections in terms of transduction efficiency across different retinal cell types, immunogenicity, procedural complexity, and disease-specific applicability. We summarized the respective advantages and limitations of each method, with a particular focus on evaluating their translational challenges and suitability in distinct disease contexts. Additionally, we explored alternative delivery techniques with potential translational advantages. These revisions aim to enhance the manuscript's analytical depth and practical relevance.

This revision can be found in lines 137-252 of the revised manuscript.

Comment No.3:

Additionally, expanding the discussion to include patient-centric outcomes could broaden the manuscript's relevance, particularly for clinicians, healthcare providers, and policy stakeholders.

Response:

Thank you for your valuable comments. Patient-centered outcome measures reflect the most intuitive improvements in visual function and quality of life brought by AAV-mediated gene therapy for retinal diseases. We fully agree with your suggestion to enrich the manuscript in this regard. Accordingly, we have revised the relevant section to explicitly incorporate patient-centered outcome evaluations. Specifically, we emphasize the importance of functional vision assessments and real-world performance metrics in IRD clinical trials, which offer greater clinical relevance than traditional structural or visual acuity endpoints. We hope these revisions enhance the clinical applicability of the manuscript.

This revision can be found in lines 281-289 of the revised manuscript.

Comment No.4:

The inclusion of commentary on ethical and regulatory aspects of retinal gene therapy, or disparities in access to clinical trials globally, would further enrich the manuscript.

Response:

We have added a new Section 7 to specifically address the ethical, regulatory, and global access challenges associated with retinal gene therapy. This section provides commentary on cost barriers, disparities in access to clinical trials between high-income and low- to middle-income countries, and underutilization of gene therapy infrastructure in primary care. It also discusses the importance of robust ethical oversight—including off-target monitoring, long-term safety evaluation, and inclusive governance—and emphasizes the need for international regulatory harmonization to prevent unequal standards and practices. We believe this addition meaningfully enriches the manuscript by offering a broader and more globally conscious perspective on translational challenges.

This revision can be found in lines 587-628 of the revised manuscript.

Overall, we have further optimized the structure of the manuscript to place greater emphasis on the multifaceted challenges faced by AAV in current clinical trials. These include technical gaps (such as large gene delivery, targeting specificity, and transduction efficiency), the potential risk of gene therapy-associated uveitis, and broader societal barriers such as disease burden. We have introduced a novel conceptual framework—the "trilemma" of precision, persistence, and practicality—as a means of evaluating the translational potential of AAV-based therapies. This addition deepens the thematic focus of the manuscript and enhances its originality. Furthermore, we have incorporated a more critical analytical perspective into previously descriptive sections and carefully addressed each suggestion aimed at improving the manuscript's innovativeness.

All revisions are clearly indicated in the main text. We sincerely thank you for your valuable comments, which have greatly improved the overall quality of our work.